# LL-VQ-VAE: Learnable Lattice Vector-Quantization For Efficient Representations

## Abstract

In this paper we introduce *learnable lattice* vector quantization and demonstrate its effectiveness for learning discrete representations. Our method, termed LL-VQ-VAE, replaces the vector quantization layer in VQ-VAE with lattice-based discretization. The learnable lattice imposes a structure over all discrete embeddings, acting as a deterrent against codebook collapse, leading to high codebook utilization. Compared to VQ-VAE, our method obtains lower reconstruction errors under the same training conditions, trains in a fraction of the time, and with a constant number of parameters (equal to the embedding dimension $D$), making it a very scalable approach. We demonstrate these results on the FFHQ-1024 dataset and include FashionMNIST and Celeb-A.

## 1 Introduction

The performance of a model heavily relies on the choice of data representation or features used during training. To ensure effective learning, extensive effort is dedicated to designing pre-processing pipelines and data transformations that can generate suitable representations of the data. Traditionally, this process depended on human creativity and domain knowledge for feature extraction. However, in order to automate this process and improve efficiency, unsupervised training is employed to automatically learn better data representations (Bengio et al., 2013; Radford et al., 2015). Many recent approaches achieve this by relying on bottleneck layers to limit data flow via compression, enabling the learning of only relevant features in the data (Yu & Seltzer, 2011; Tishby et al., 2000). One of the paramount examples are variational autoencoders (VAE) that achieve this by compressing data into lower-dimensional latent variables represented by Gaussian distributions (Takida et al., 2022).

Many applications require further control over the learned features, constraining them to a finite set of representations, through discretizing the latent space. The most notable discrete latent model is the Vector-Quantized Variational Auto-Encoder (VQ-VAE) Van Den Oord et al. (2017), which many recent works successfully leveraged to train language models over continuous data (Yan et al., 2021; Bao et al., 2021). The traditional VQ-VAE uses vector quantization to learn a $K$-sized codebook of discrete latent variables by training an approximation of online-$k$ means clustering, employing a pass-through estimator to approximate the gradient Van Den Oord et al. (2017). There are two approaches for updating the codebook: (1) minimizing the Mean Squared Error (MSE) based on the encoder latents, (2) utilizing an Exponential Moving Average (EMA) over the codebook. The first method is widely preferred but suffers from codebook collapse, where the latents become quantized to only a small subset of all available options (Łańcucki et al., 2020; Takida et al., 2022). Additionally, the computational complexity of this alternative scales with the size $K$ of the codebook. On the other hand, the second option is significantly faster, although it does not achieve the same level of quantization, resulting in much larger codebooks and higher reconstruction errors. Both methods are commonly utilized in online implementations, forcing users to choose what to prioritize: quantization vs. speed. These problems motivate us to explore alternative quantization techniques in order to improve model efficiency, avoiding codebook collapse, while providing users with high quality quantizations without sacrificing speed.

Lattice quantization is a variant of vector quantization which utilizes a regular lattice structure to represent embeddings (Gibson & Sayood, 1988). Unlike typical vector quantization that relies on arbitrary sets of representative vectors, lattices offer a systematic arrangement of points in space,

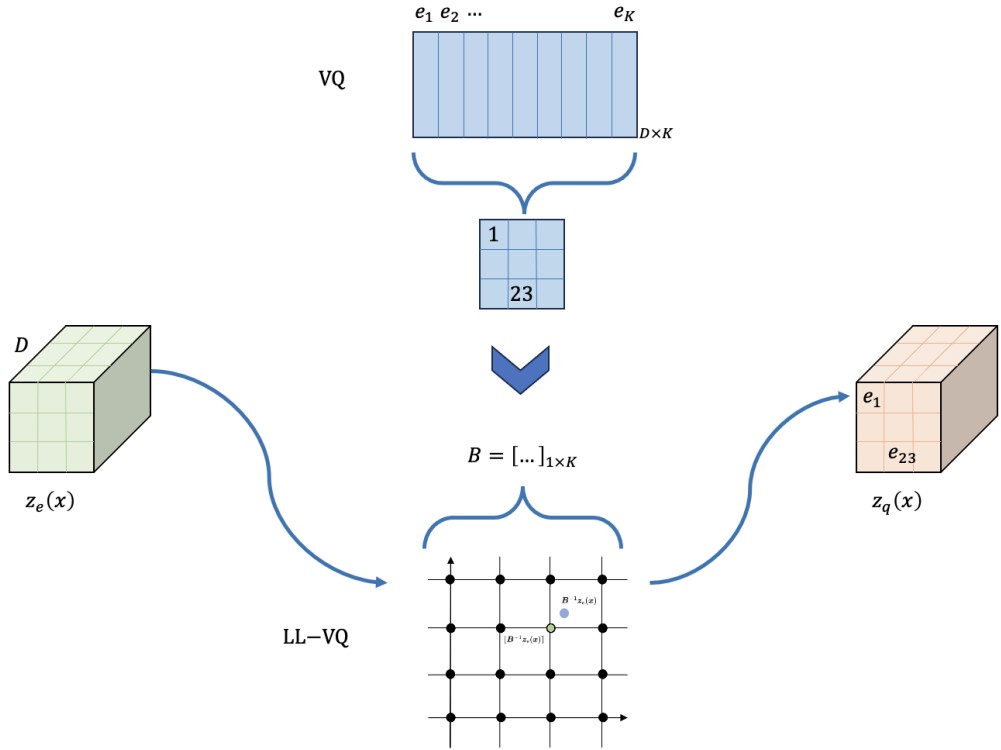

Figure 1: System overview. We replace vector quantization (VQ) with learnable lattice vector quantization (LL-VQ). The VQ layer learns each embedding vector by training $D \times K$ parameters while the LL-VQ layer only learns the $K$ parameters defining the lattice.

enhancing the representation of embeddings through discrete mathematical structures. This ordering simplifies the process of quantizing to the closest vector under certain conditions (Agrell et al., 2002). For example, several works use lattice quantization schemes for efficient compression of the decentralized model updates in federated learning (Shlezinger et al., 2020; Zong et al., 2021; Zhang & Zhang, 2023). Choi et al. (2020) exploits lattices to quantize the weights of deep neural networks and produce memory-efficient compressions, while Zhang & Wu (2023) replace scalar quantizers with a pre-defined lattice quantizer to build an end-to-end image compression system, obtaining better rate-distortion performance but with a non-significant increase in model complexity.

In this paper we use lattice quantization for efficient latent discretization (Figure 1). Our contributions can be summarized as follows. We

- introduce the Learnable Lattice VQ-VAE (LL-VQ-VAE), which replaces vector quantization with a *learnable* lattice layer for discretizing latent variables.

- describe and demonstrate the main practical properties and considerations of the approach, including the natural aversion of lattice quantization to codebook collapse due to the imposed structure on the embeddings, reducing the likelihood of any being favored over others. We significantly reduce the number of training parameters in the quantization layer no matter the desired codebook size $K$.

- report high quantization speeds without sacrificing quantization quality, providing users with both without having to surrender either. Empirically, we show the superiority of our reconstructions across different challenging datasets like FFHQ-1024 and Celeb-A.

## 2 BACKGROUND

VQ-VAEs (Van Den Oord et al., 2017) represent high-dimensional $D$ input data $\boldsymbol{x}$ with a finite $K$-sized set of discrete low-dimensional embedding vectors $\boldsymbol{z}$. The model consists of three components: a decoder network parameterizing the distribution $p(\boldsymbol{x}|\boldsymbol{z})$, a quantization layer over a uniform prior $p(\boldsymbol{z})$, and encoder network with a categorical posterior distribution approximated as:

$$q(\boldsymbol{z} = k|\boldsymbol{x}) = \begin{cases} 1 & \text{for } k = \text{argmin}_i \|\boldsymbol{z}_e(\boldsymbol{x}) - \boldsymbol{e}_i\|_2 \\ 0 & \text{otherwise} \end{cases} \tag{1}$$

Vector quantization is used to map the encoder output $\boldsymbol{z}_e(\boldsymbol{x})$ to the nearest discrete code $\boldsymbol{e}_i$ in the $K$-sized codebook $(e_i)_{i=1}^K$. All three layers are trained conjointly using the following training objective:

$$\log p(\boldsymbol{x}|\boldsymbol{z}_q(x)) + \|\text{sg}[\boldsymbol{z}_e(\boldsymbol{x})] - \boldsymbol{e}\|_2^2 + \beta\|\boldsymbol{z}_e(\boldsymbol{x}) - \text{sg}[\boldsymbol{e}]]\|_2^2, \tag{2}$$

where "sg" denotes the stop gradient operator. The first term is the *reconstruction error* and is used to train the encoder and decoder. The second term, *embedding loss*, trains the quantization layer but is often substituted with an online $k$-means clustering exponential moving average update. The third term is the *commitment loss* and is used to constrain the encoder outputs from growing arbitrarily.

## 3 METHODOLOGY

### 3.1 LATTICE QUANTIZATION

We define a learnable lattice $\mathcal{L}(\boldsymbol{B}) = \{\boldsymbol{B}\boldsymbol{v} : \boldsymbol{v} \in \mathbb{Z}^D\}$, where $\boldsymbol{B}$ is the lattice basis matrix, $\boldsymbol{v}$ is an integer vector, and $D$ is the space dimensionality. The encoder takes in an image $\boldsymbol{x}$ and outputs an embedding $\boldsymbol{z}_e(\boldsymbol{x})$, which $\boldsymbol{z}_e(\boldsymbol{x})$ is quantized to the nearest lattice point $\boldsymbol{e}_i$ using the Babai Rounding Estimate (BRE) (Equation 4). We define an approximate categorical posterior over $\mathbb{Z}$:

$$q(\boldsymbol{z} = \boldsymbol{v}_i|\boldsymbol{x}_i) = \begin{cases} 1 & \text{for } \boldsymbol{v}_i = \lfloor \boldsymbol{B}^{-1}\boldsymbol{z}_e(\boldsymbol{x}_i) \rceil \\ 0 & \text{otherwise} \end{cases} \tag{3}$$

The integer lattice $\mathbb{Z}$ is analogous to the codebook defined in VQ-VAE, where each point $\boldsymbol{v}_i$ on the lattice is treated as a unique latent code index.

$$\boldsymbol{e}_i = \boldsymbol{B}\boldsymbol{v}_i = \boldsymbol{B}\lfloor \boldsymbol{B}^{-1}\boldsymbol{z}_e(\boldsymbol{x}_i) \rceil, \tag{4}$$

The BRE is an approximation to the Closest Vector Problem, meaning $\boldsymbol{e}_i$ isn't guaranteed to be the closest lattice point to $\boldsymbol{z}_e(\boldsymbol{x}_i)$ but one that is close enough. We remedy this by defining $\boldsymbol{B}$ to be a diagonal matrix (equation 5), making the lattice basis linearly independent and guaranteeing that the BRE would indeed find the closest lattice point $\boldsymbol{e}_i$ to the embedding vector $\boldsymbol{z}_e(\boldsymbol{x}_i)$ (Agrell et al., 2002). Figure 2 shows a 2-dimensional example of lattice quantization.

$$\boldsymbol{B} = \begin{cases} b_{ij}, & \text{if } i = j \\ 0, & \text{if } i \neq j \end{cases} \tag{5}$$

### 3.2 CONSTRAINING THE LATTICE

Since $\mathcal{L}(\boldsymbol{B})$ spans $\mathbb{Z}^D$, our codebook is effectively of infinite size. We found that without further constraints our quantization layer would quantize each embedding vector to its own unique point on the lattice. Whilst this results in higher-quality reconstructions, it defeats our main objective of discretizing the embedding space into a small, finite set of embedding vectors. Therefore, to produce a desired codebook size $K$ we apply two techniques in constraining the lattice. First, we set the initial lattice sparsity, which directly affects the resulting codebook size, by uniformly initializing $\boldsymbol{B}$

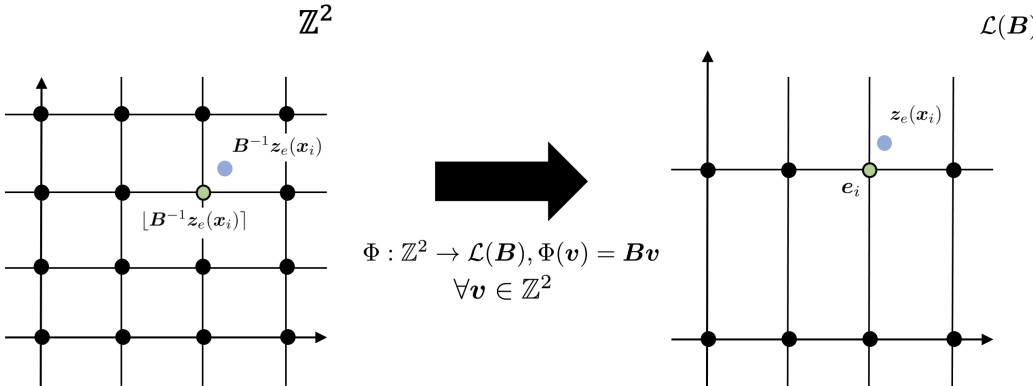

Figure 2: Quantization on a 2-dimensional lattice. The discrete latent vectors are the result of linearly transforming the integer domain $\mathbb{Z}^2$ using the basis matrix $\boldsymbol{B}$. To obtain the index of the nearest lattice vector to an encoder embedding we merely apply the inverse transformation to $\boldsymbol{z}_e(\boldsymbol{x}_i)$ then round to the nearest integer. This simple process works since $\boldsymbol{B}$ is a diagonal matrix.

as in equation 6 (derived in appendix A.1). This range is derived by assuming idealized quantization on a set of linearly independent dimensions, providing us with a starting point for the lattice density.

$$\boldsymbol{B} \sim U\left(-\frac{1}{\sqrt[D]{K} - 1}, \frac{1}{\sqrt[D]{K} - 1}\right). \tag{6}$$

Second, we push the lattice towards increased sparsity by adding a size loss term (equation 7) to the training objective, which increases the basis determinant resulting in greater spacing between the lattice points in a given bounded region,

$$-\gamma \|\text{diag}(\boldsymbol{B})\|_1, \tag{7}$$

where $\gamma$ is the sparsity coefficient and can be used to either scale the sparsity constraint or completely reverse it. By setting $\gamma$ to $-1$ we push the lattice to be as dense as possible, effectively quantizing the data to a codebook of infinite size.

The total training objective becomes:

$$\log p(\boldsymbol{x}|\boldsymbol{z}_q(x)) + \|\text{sg}[\boldsymbol{z}_e(\boldsymbol{x})] - e\|_2^2 + \beta \|\boldsymbol{z}_e(\boldsymbol{x}) - \text{sg}[e]\|_2^2 - \gamma \|\text{diag}(\boldsymbol{B})\|_1, \tag{8}$$

where $\boldsymbol{z}_q(x)$ is the decoder input and $\beta$ is the commitment cost. The first three terms of the objective are identical to that of the VQ-VAE.

### 3.3 PRACTICAL CONSIDERATIONS AND LIMITATIONS

**Scalability.** The LL-VQ-VAE's size and computational complexity are completely agnostic to the desired number of embeddings $K$. This makes our method scalable with any desired codebook size. Furthermore, we need only keep track of $D$ parameters instead of $D \times D$ since $\boldsymbol{B}$ is a diagonal matrix and therefore not fully utilized.

**Uniformity and regularization.** As opposed to vector quantization, where each code is it independent from the others, all points on the lattice are intrinsically coupled by the underlying lattice structure. This ensures the latent codes are uniformly distributed across the embedding space, meaning no areas are more dense/sparse over others. It further acts as a regularizer over the codebook as moving one latent code means moving the entire lattice.

Table 1: Key differences between lattice and vector quantization. Lattice quantization uses less trainable parameters, has high aversion to codebook collapse due to the underlying structure, and does not scale in complexity with the desired codebook size.

| Quantization method | Layer size | Quantization complexity | Aversion to codebook collapse |
|---|---|---|---|
| Vector | $K \times D$ | $O(K)$ | Low |
| Lattice | $D$ | $O(1)$ | High |

**Codebook collapse.** The combined result of those two effects mentioned above reduces the likelihood of codebook collapse, a common problem with VQ-VAEs (Łańcucki et al., 2020; Takida et al., 2022). In fact, we found that without any constraining the lattice is always driven to an increased density, demonstrating that the LL-VQ-VAE has natural disinclination towards codebook collapse.

**Upper limit on $K$.** Since there is no upper limit on the number of points on a lattice, our quantization layer has vast flexibility in controlling the number of embeddings $K$ as driven by the training objective. However, this also means that, unlike the VQ-VAE, we cannot impose an upper limit on $K$ but only drive the lattice towards a desired $K$ through several techniques as detailed in Section 3.2.

Table 1 contains a summary of property comparisons between lattice and vector quantization.

## 4 EXPERIMENTS

We conduct experiments illustrating the differences between lattice and vector quantization on the FFHQ-1024 dataset. Further results on FashionMNIST and CELEB-A can be found in Appendix A.2. We also include experiments on the lattice initialization, structure, and sparsity.

### 4.1 ARCHITECTURE AND PARAMETERS

All models use the same encoder and decoder architectures. The encoder consists of 6 layers with a LeakyReLU activation function appended to each one: 2 convolutional layers of hidden dimensions 16 and 32 respectively (kernel size 4, stride 2, and padding 1), 1 convolutional layer of 32 hidden units (kernel size 3, stride 1, and padding 1), 2 residual layers, and a final convolutional layer 32 hidden units (kernel size 1 and stride 1). Similarly, the decoder is 6 layers with a LeakyReLU activation function appended to each one: a convolutional layer of 32 hidden units (kernel size 3, stride 1, and padding 1), 2 residual layers, 2 convolutional layers of hidden dimensions 32 and 16 respectively (kernel size 4, stride 2, and padding 1), and a final convolutional layer 16 hidden units (kernel size 4, stride 2, and padding 1). The residual layers are implemented as Conv2D (kernel size 3, padding 1, and no bias) followed by ReLU followed by Conv2D (kernel size 1, padding 1, and no bias).

Training was performed for 5 epochs on an NVIDIA GeForce RTX 4090 GPU with a 32 batch size, 0.001 learning rate, exponential learning rate scheduler with 0.0 gamma, commitment cost 0.25, embedding dimension $D = 64$, and number of embeddings $K = 512$. The reported results are the aggregate of 3 random seeds.

### 4.2 COMPARISON WITH VQ-VAE

We compare the VQ-VAE and LL-VQ-VAE on the FFHQ-1024 dataset in Table 2. We also include the EMA updated version, which was first mentioned in the appendix of Van Den Oord et al. (2017). Results demonstrate that the LL-VQ-VAE obtains lower reconstruction errors than both VQ-VAE variants under the same model architecture and training parameters (Figure 3).

Using a lattice imposes a uniform structure on $\mathbb{R}^D$ by mapping the integer domain $\mathbb{Z}^D$; therefore, we can easily infer the index of any embedding vector given we know the linear mapping $\boldsymbol{B}$. This property leads to a significant reduction in the number of learning parameters needed to train the

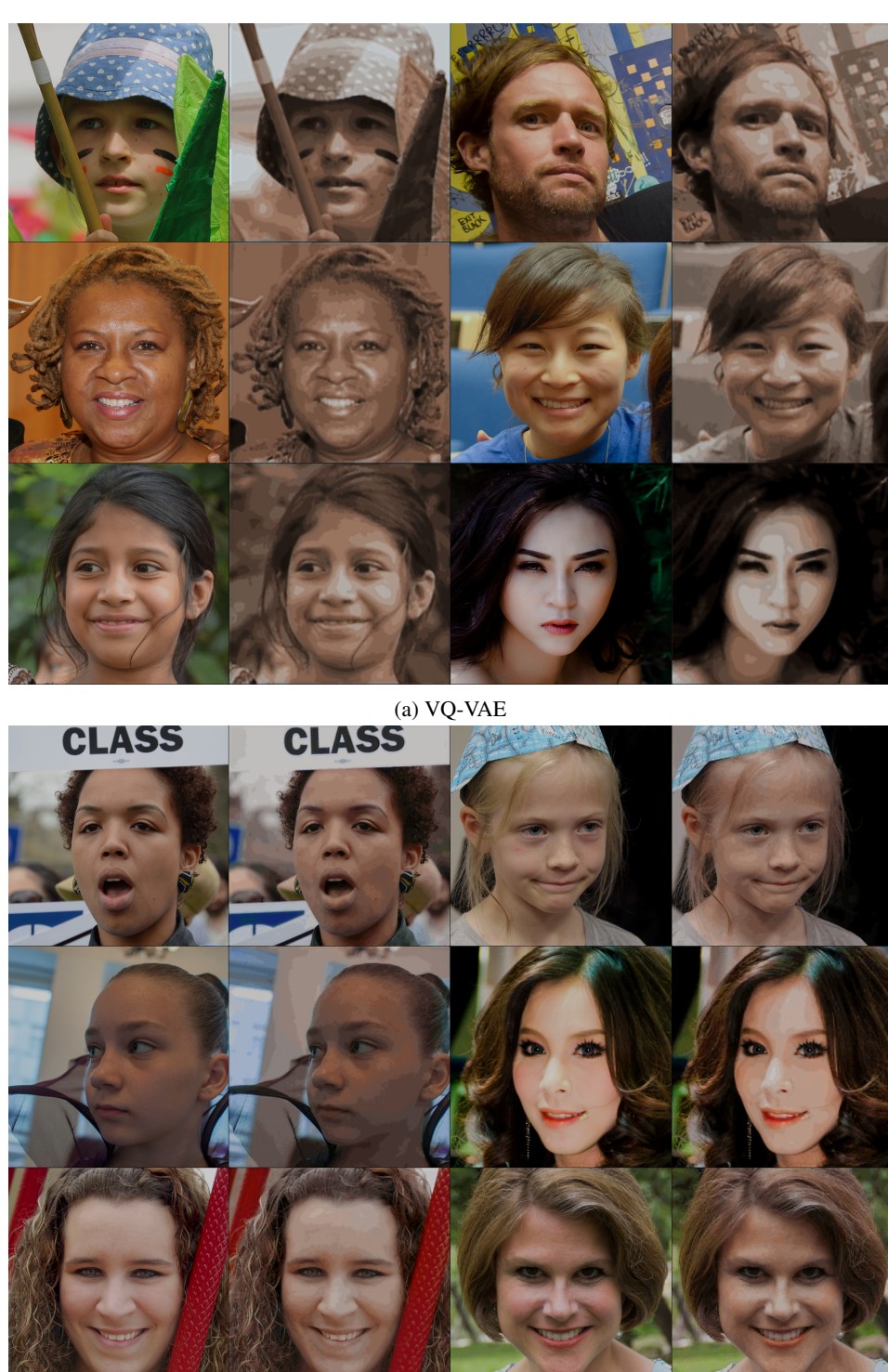

(a) VQ-VAE

(b) LL-VQ-VAE

Figure 3: FFHQ-1024 reconstructions. The LL-VQ-VAE obtains higher quality reconstructions than the VQ-VAE, under the same network architecture and training parameters.

Table 2: Results comparing VQ-VAE, VQ-VAE (EMA), and LL-VQ-VAE on the FFHQ-1024 dataset quantization. The LL-VQ-VAE obtains the lowest reconstruction error, is faster than either method, does not suffer from codebook collapse nor explosion, and only has $D = 64$ training parameters.

| Model | Layer size | Recon. error | No. embeddings/dataset | Duration (mins) |
|-------|-----------|--------------|------------------------|-----------------|
| VQ-VAE | 32,768 | 0.0063 | 36 | 231 |
| VQ-VAE (EMA) | 65,536 | 0.0033 | 87,982 | 85 |
| LL-VQ-VAE | **64** | **0.0018** | **1,405** | **79** |

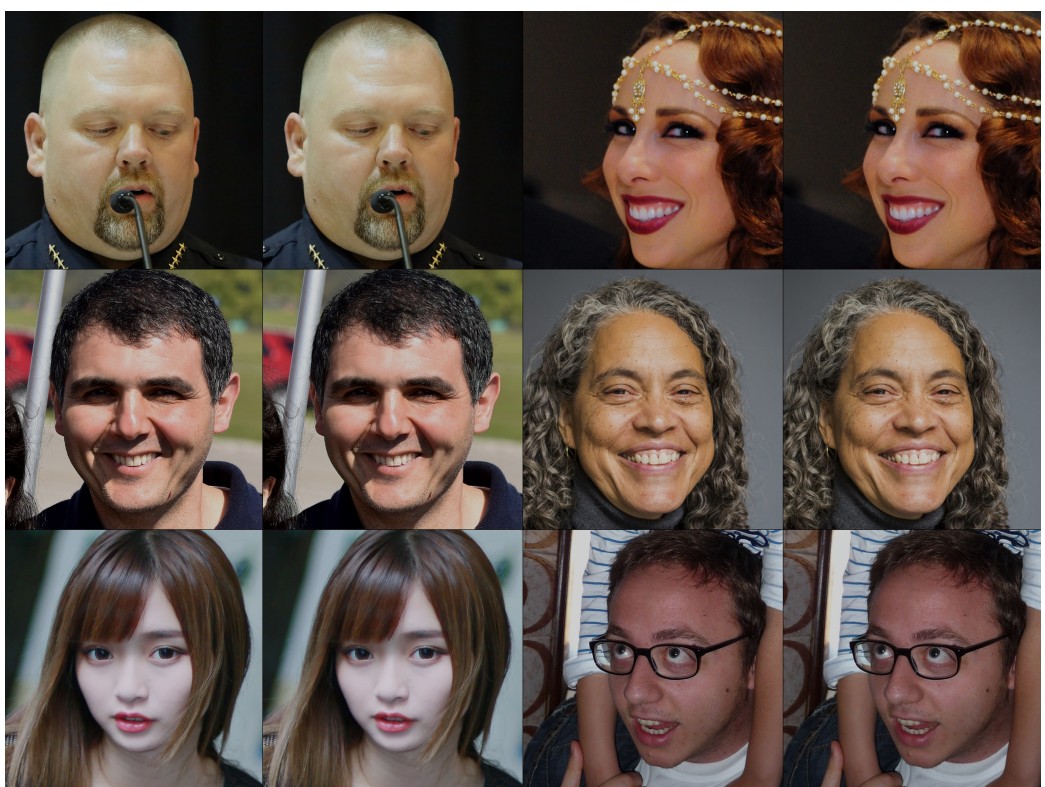

Figure 4: The lattice learns a very dense structure resulting in high-quality reconstructions without any added computational complexity.

lattice quantizer. As seen in Table 2, the LL-VQ-VAE only learns 64 parameters in the quantization layer as opposed to both VQ-VAE variants, where the VQ-VAE and VQ-VAE (EMA) learn $\frac{32,768}{64} = 512$ and $\frac{65,536}{64} = 1,024$ times the LL-VQ-VAE's number of parameters, respectively.

The usage of a lattice structure further simplifies the quantization technique, decoupling the computation complexity from the scale of the codebook size. This is demonstrated by the LL-VQ-VAE's short training time, which is a fraction of the time taken by the VQ-VAE $\frac{84}{235} = 0.36$ and is as fast as the VQ-VAE (EMA) $\frac{84}{86} = 0.98$.

Table 2 further exhibits the LL-VQ-VAE's aversion to codebook collapse/explosion by obtaining a number of embeddings/dataset $1,405$ close enough to the desired number of embeddings $K = 512$ without being too large (codebook explosion) nor too small (codebook collapse). The VQ-VAE on the other hand obtains a low number of 36 embeddings/dataset, showing that even with a hard limit on the codebook size, vector quantization naturally collapses the codebook to a select few embeddings. The lattice however couples all embeddings to one another so as one embedding vector moves towards an encoder embedding, the entire lattice moves as well. This coupling acts as a

Table 3: Lattice initialization and density have direct impact over the resulting codebook size and reconstruction quality. As the lattice learns more dense structures, it obtains lower reconstruction errors.

| Init. range | Sparsity coef. | Target $K$ | Recon. error | No. embeddings/dataset |
|---|---|---|---|---|
| (-1, 1) | 1 | 1.84e19 | 0.0016 | 7,439 |
| (-9.76, 9.76) | 1 | 512 | 0.0018 | **1,405** |
| (-1, 1) | -1 | - | **0.0006** | - |

regularization technique against the encoder honing in on a select few latent vectors, preventing codebook collapse.

The VQ-VAE (EMA) suffers from the opposite problem to codebook collapse as it results in a very high number of embeddings/dataset $87, 982$, effectively defeating the main objective of latent discretization. We believe this technique is widely found in VQ-VAE implementations due to its better reconstructions and shorter training time as opposed to the vanilla VQ-VAE. The choice is usually up to the discretion of the user based on what they prioritize: quantization vs. speed. Our method provides both without sacrificing either. In short, the LL-VQ-VAE: obtains lower reconstruction errors than either VQ-VAE variant; is even faster than the EMA variant; outputs a codebook size close to the desired number without codebook collapse nor explosion; and is very small, training $D$ parameters only.

### 4.3 LATTICE INITIALIZATION & SPARSITY

We showcase the effect of lattice initialization on the resulting lattice sparsity. Table 3 includes 2 lattices with sparsity coefficient 1 but different target $K$s. Results demonstrate that even with a vastly dense initial lattice the LL-VQ-VAE will still produce a relatively sparse structure due to the size loss term.

There is a clear inverse correlation between the resulting codebook size $K$ and reconstruction error; the more dense the lattice the better the reconstructions. This is demonstrated by setting the sparsity coefficient to $-1$, effectively flipping the sparsity loss term and pushing the lattice to be as dense as possible.

Figure 4 clearly shows the superior reconstructions obtained by the dense lattice; however a the cost of no discretization. We do not report the number of embeddings/dataset for the dense lattice in Table 3 as it is not feasible to obtain.

## 5 RELATED WORK

In this work we present lattice quantization (Gibson & Sayood, 1988) and use it for learning discrete latent variables (Mnih & Gregor, 2014) in variational autoencoders (Kingma & Welling, 2013; Rezende et al., 2014). Discretizing the latent space has had powerful impact in recent years in multiple disciplines such as image generation (Yu et al., 2022; Chen et al., 2020; Ho et al., 2022), speech recognition (Baevski et al., 2020), and reinforcement learning (Janner et al., 2021).

**VQ-VAE and extensions.** There exists multiple approaches to learning discrete latent variables in VAEs such as NVIL (Mnih & Gregor, 2014), VIMCO (Mnih & Rezende, 2016), Concrete (Maddison et al., 2016) and Gumbel-softmax (Jang et al., 2016) based methods. However the most prominent approach is the VQ-VAE (Van Den Oord et al., 2017) which was the first to tackle complex datasets such as CIFAR10, ImageNet, and DeepMind Lab, and a raw speech dataset (VCTK) and obtain performances comparative to continuous latent variable VAEs. Furthermore, the VQ-VAE paper demonstrated the usage of discrete codebooks to train autoregressive priors like PixelCNN (Van Den Oord et al., 2016) and WaveNet (Oord et al., 2016). Later approaches expanded the VQ-VAE into hierarchical frameworks (Williams et al., 2020), adding more capacity to the quantization layer without significant increases to the desired number of embeddings, demonstrating competitive results against BigGAN (Razavi et al., 2019).

**Lattices and representation learning.** Recent works attempt to combine lattice quantization with VAEs. Lastras (2020) introduces a form of lattice quantization where they use additive dither noise and a lower bound on the training objective through a prior to learn lattice embeddings. However, they do not demonstrate the efficiency and practicality of lattice quantization as they only use non-finite lattices, negating the main objective of discretizing the latent space. Furthermore, they only experiment on simple datasets like MNIST and OMNIGLOT with no conclusive results on their reconstruction quality as opposed to VQ-VAE. Kudo et al. (2022) introduce LVQ-VAE demonstrating SOTA rate-distortion performance by utilizing a constrained codebook and jointly optimizing a distribution over the lattice (known as the entropy model) with a hyperprior and spatially autoregressive model. However the LVQ-VAE suffers from very high complexity and slow run-time, rendering the model impractical. Neither work introduce the notion of a learnable lattice, but instead rely on fixed basis matrices during training.

Our work introduces a very simple and efficient approach for learnable lattice vector quantization, outperforming VQ-VAE in quality, memory, and complexity on datasets like FFHQ-1024 and Celeb-A. Our solution further provides users with the flexibility of choosing quality over quantization by altering the sparsity coefficient, affecting the lattice density.

## 6    CONCLUSION

In this paper we introduce the LL-VQ-VAE which replaces vector quantization in the VQ-VAE with a *learnable lattice*. The training objective pushes the lattice into a structure that balances reconstruction quality with codebook size. All discrete latents on the lattice are coupled leading to various useful properties: (1) fast and efficient quantization, (2) low number of trainable parameters that does not scale with the desired codebook size, (3) regularization against favouring certain embeddings over others, preventing codebook collapse.

When compared to vector quantization, our method exhibits higher quality reconstructions on all tested datasets. We demonstrate this on the FFHQ-1024 dataset and include FashionMNIST and Celeb-A in Appendix A.2. The LL-VQ-VAE provides users with high-quality latent discretization without sacrificing computational complexity as a trade-off.

In the future we would like to further explore how the different quantization strategies are linked to preserving low- and mid-level image properties, such as contrast or brightness, and how effective the representations are in terms of resilience against distortions.

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

## A  APPENDIX

### A.1  DERIVATION OF THE LATTICE BASIS INITIALIZATION RANGE

For a lattice $\mathcal{L}(\boldsymbol{B})$, assume that each dimension $d$ only exists in the range $[0, 1]$. Therefore counting the number of lattice points per dimension $k_i$ is simply:

$$k_i = \frac{1}{b_{j,l}} + 1, \tag{9}$$

where $i = j = l$ since $\boldsymbol{B}$ is a diagonal matrix. So the total number of lattice points $K$ becomes:

$$K = \prod_{i=1}^{D} k_i + 1 = \left( \frac{1}{b_{j,l}} + 1 \right)^D \tag{10}$$

We can rearrange this equation to obtain an expression for the $\boldsymbol{B}$ diagonal values that would result in $K$ lattice points:

$$b_{j,l} = -\frac{1}{\sqrt[D]{K} - 1} \tag{11}$$

### A.2  FURTHER QUANTIZATION RESULTS

All models use the same encoder and decoder architectures. The encoder consists of 5 layers with a LeakyReLU activation function appended to each one: 1 convolutional layer of hidden dimensions 64 respectively (kernel size 4, stride 2, and padding 1), 1 convolutional layer of 64 hidden units (kernel size 3, stride 1, and padding 1), 2 residual layers, and a final convolutional layer 64 hidden units (kernel size 1 and stride 1). Similarly, the decoder is 5 layers with a LeakyReLU activation function appended to each one: a convolutional layer of 64 hidden units (kernel size 3, stride 1, and padding 1), 2 residual layers, 1 convolutional layers of hidden dimensions 64 (kernel size 4, stride 2,

Table 4: Comparisons on Celeb-A quantization. The patterns here are identical to those of the FFHQ-1024 quantization results with the exception of VQ-VAE (EMA) obtaining worse reconstructions than the vanilla VQ-VAE.

| Model | Recon. error | No. embeddings/dataset | Duration (mins) |
|---|---|---|---|
| VQ-VAE | 0.0025 | 98 | 54 |
| VQ-VAE (EMA) | 0.0057 | 34,210 | 19 |
| LL-VQ-VAE | 0.0017 | **890** | 17 |
| LL-VQ-VAE (dense) | **0.00009** | - | **16** |

Table 5: Comparisons on Fashion-MNIST quantization. We note that given the simplicity of the data, there is no drastic difference in quantization speed between all methods. However, the same patterns in reconstruction quality and codebook size as with other datasets hold.

| Model | Recon. error | No. embeddings/dataset | Duration (secs) |
|---|---|---|---|
| VQ-VAE | 0.013 | 32 | 89 |
| VQ-VAE (EMA) | 0.047 | 63,566 | **84** |
| LL-VQ-VAE | 0.009 | **1,781** | **84** |
| LL-VQ-VAE (dense) | **0.007** | - | 85 |

and padding 1), and a final convolutional layer 16 hidden units (kernel size 4, stride 2, and padding 1). The residual layers are implemented as Conv2D (kernel size 3, padding 1, and no bias) followed by ReLU followed by Conv2D (kernel size 1, padding 1, and no bias).

Training was performed for 5 epochs on an NVIDIA GeForce RTX 4090 GPU with a 64 batch size, 0.001 learning rate, exponential learning rate scheduler with 0.0 gamma, commitment cost 0.25, embedding dimension $D = 64$, and number of embeddings $K = 512$. The reported results are the aggregate of 3 random seeds.

We note that the VQ-VAE (EMA) does not always obtain better reconstructions but always an explosion of codebook size when compared to the VQ-VAE. This is the case in both Tables 4 and 5. Figure 5 and 6 shows Celeb-A and Fashion-MNIST reconstructions respectively.

Both tables show the LL-VQ-VAEs aversion to codebook collapse, but Table 5 shows that the learnable lattice could choose to increase its density in favor for reconstruction quality. This is likely due to the model architecture with respect to the data complexity.

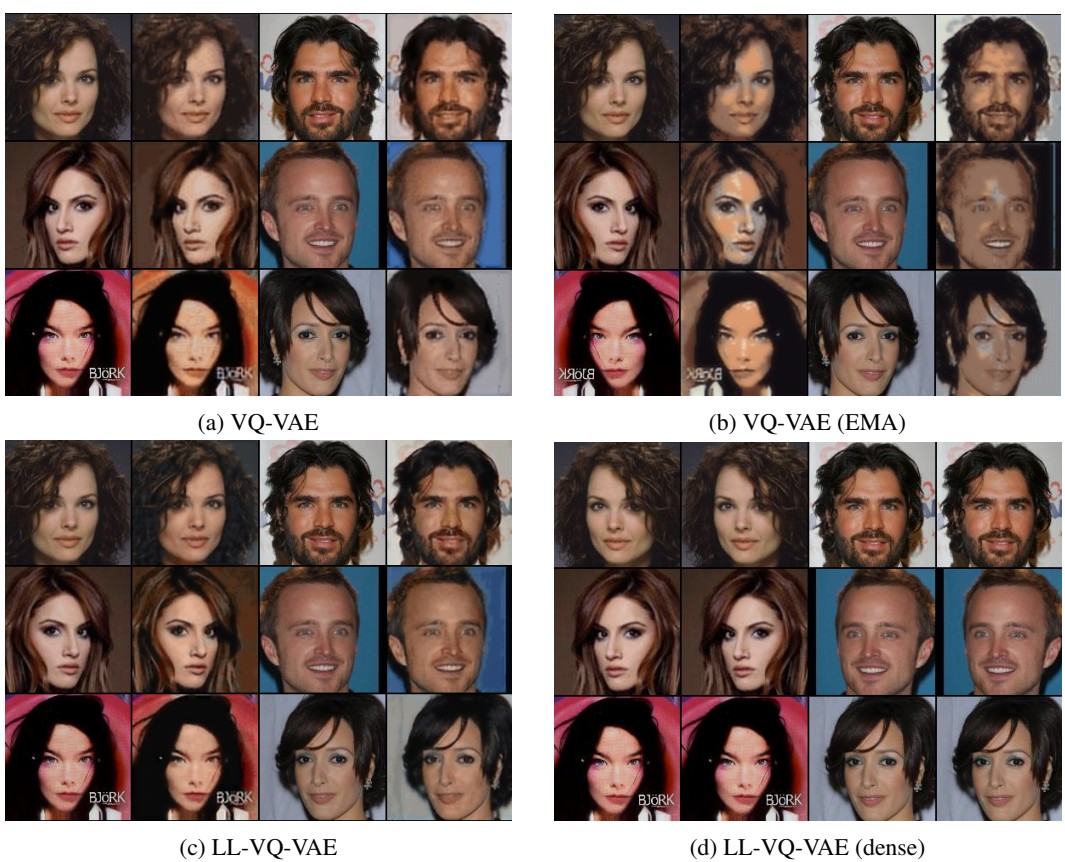

(a) VQ-VAE

(b) VQ-VAE (EMA)

(c) LL-VQ-VAE

(d) LL-VQ-VAE (dense)

Figure 5: Celeb-A sample reconstructions

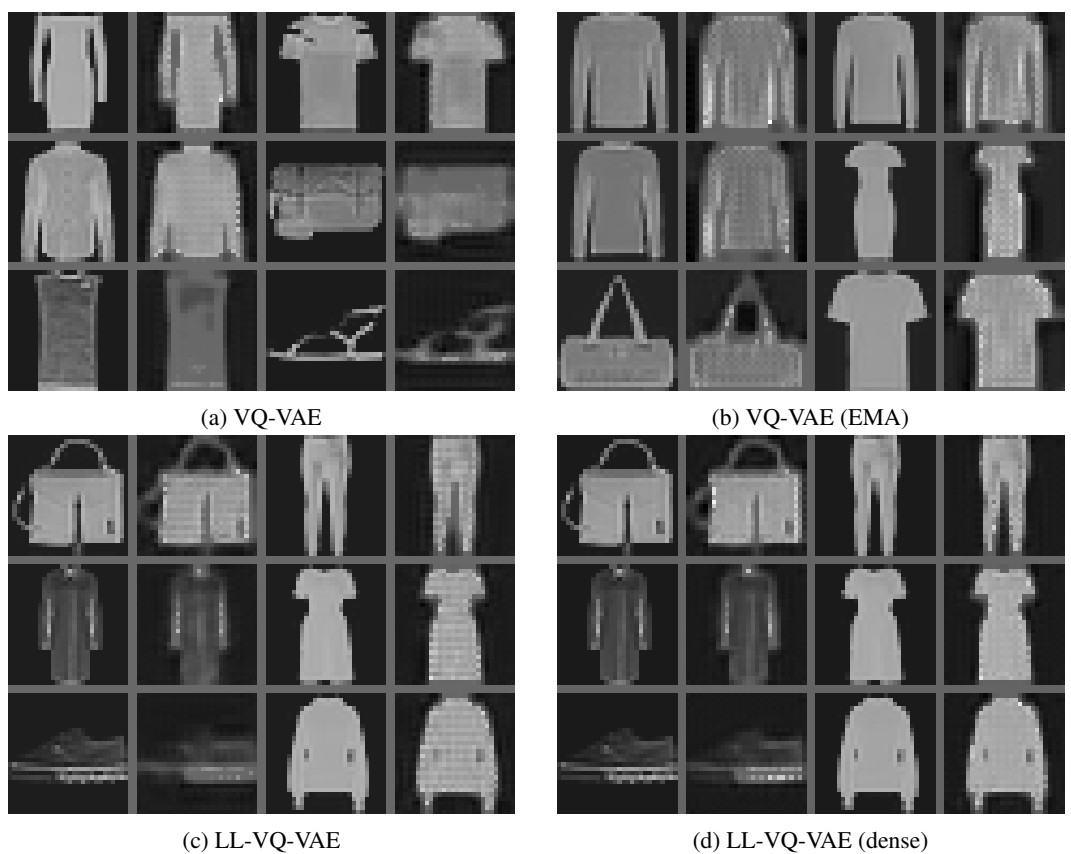

Figure 6: Fashion-MNIST sample reconstructions

