# OpenReview forum: "LL-VQ-VAE: Learnable Lattice Vector-Quantization For Efficient Representations"
_ICLR.cc/2024/Conference — Submitted to ICLR 2024_

### Official Review · Reviewer_yCAH · 2023-11-01

**Soundness:** 3 good
**Presentation:** 2 fair
**Contribution:** 3 good
**Rating:** 5
**Confidence:** 2

**Summary:**

Propose a learnable lattice quantization method for use in VQ-VAE models. The lattice is used in the categorical posterior, and is sparsity constrained. Results are presented on FFHQ-1024, FahsionMNIST, and Celeb-A.

**Strengths:**

- To the best that I can tell, the proposed method seems to be an interesting and well reasoned extension to the previous VQ-VAE model. I think that using a learned lattice is a clever proposal.
- The results seem to be quite impressive, but I am unfamiliar with the VAE literature and confused enough by the writing that I could be mistaken.

**Weaknesses:**

While I think there is a good idea behind the paper, I find the writing and presentation of the method to be confusing. I am not familiar with VAEs, but I am familiar with some deep learning quantization works (and have published some at neurips). I feel that someone of my background should still be able to understand the paper. For example, in Section 3.2 why does enforcing sparsity in the lattice ensure shared points? Also, can the authors explain what is the evidence for resistance to codebook collapse? I see that the no embeddings column is close to the desired 512, but I think that more than one data point is required when making an inference about empirical trends. I also am confused by what Fig1 is supposed to be showing. It could help if the various aspects (D, VQ, z_q, etc) are explained.

In terms of organization, I think there need to be more empirical results in the main body of the paper. Either that, or the presentation of the method could be expanded.

**Questions:**

Misc
- A minor point: a table of figure would better explain the architecture details, Section 4.1
- A personal opinion: I don’t think the figures need to be so big. I would rather have smaller figures and have more explanation on the method, or more experimental results.
- It says training was performed for 5 epochs, is this a standard amount? Again, I am not familiar with VAEs but this seems low.

---

### Official Review · Reviewer_4UQJ · 2023-11-01

**Soundness:** 3 good
**Presentation:** 2 fair
**Contribution:** 3 good
**Rating:** 6
**Confidence:** 4

**Summary:**

The proposed method, LLVQ-VAE, modifies VQ-VAE by integrating lattice-based discretization in place of the standard vector quantization layer. This learnable lattice structure helps prevent codebook collapse, ensuring better codebook utilization. The method's performance is validated on the FFHQ-1024 dataset, as well as the FashionMNIST and Celeb-A datasets.

Overall, the proposed method is straightforward yet shows promise in enhancing the performance of VQ-VAE. However, the paper would benefit from clearer writing and additional experimental evidence to reinforce its conclusions.

**Strengths:**

The authors propose a novel method for vector quantization in VAE.

Experiments on several datasets demonstrate the effectiveness and efficiency of the proposed method.

**Weaknesses:**

Table 2 presents quantitative results for only one dataset. Conducting additional quantitative experiments could more clearly highlight the pros and cons of the proposed method.

In Figures 3 and 4, are the original images displayed on the left and their reconstructions on the right?

The reconstruction results are already given in Figure 3, it is unclear why FIgure 4 is necessary.

For a clearer comparison, it would be helpful to display reconstructions of the same original image when comparing the proposed method to VQ-VAE.

I'm unclear about Table 3. What does "init. range" refer to? And does "sparsity coef." correspond to γ in Equation (8)?

**Questions:**

1. In Figures 3 and 4, are the original images displayed on the left and their reconstructions on the right?

2. Sice the reconstruction results are already given in Figure 3, why is FIgure 4 necessary?

3. What does "init. range" refer to in Table 3? And does "sparsity coef." correspond to γ in Equation (8)?

---

### Official Review · Reviewer_qZdA · 2023-11-02

**Soundness:** 2 fair
**Presentation:** 2 fair
**Contribution:** 2 fair
**Rating:** 5
**Confidence:** 2

**Summary:**

This work considers replacing the quantization layer of vector quantized variational autoencoders (VQ-VAE) with a learnable transform quantization strategy. The goal of VQ-VAE is discrete representation learning, i.e., we want to embed a datapoint in the latent space using an encoder network, and we want the representation in the latent space to belong to a discrete set of finite values (referred to as the codebook). This codebook is learnt by jointly training the encoder and decoder network.

Traditional VQ-VAE approaches consists of a quantization layer, in which the codebook (or the quantization points) is done using K-means clustering or exponential moving average. These approaches have some issues such as codebook collapse, in which the learnt codebook is a very small set of points compared to the latent space. The alternative strategy proposed in this paper, learns a transform quantization strategy, in which the transform coefficients are learnt in the training process. Doing so circumvents some problems associated with the traditional VQ-VAE approach such as codebook collapse. Furthermore, the authors also constrain the basis to be diagonal so as to learn few parameters and perform the inversion easily.

Please correct me if I am mistaken in my understanding of the contributions of the paper. I am more than happy to rectify my review if that's the case.

**Strengths:**

The authors propose an alternative discretization strategy for discrete representation learning using variational autoencoders, and show some benefits over the existing strategies.The claims are validated via numerical simulations.

The proposed strategy learns fewer parameters than traditional VQ-VAE, and hence, is computationally efficient. The complexity of doing the quantization is less compared to a traditional nearest neighbor search, because of the additional cubic structure imposed on the quantization lattice, as a consequence of which, Babai's rounding strategy can be easily applied. The proposed strategy also has a a smaller reconstruction error between the input and the decoder output, while simultaneously circumventing issues like codebook collapse.

**Weaknesses:**

I have some concerns about the paper in general. I would be happy to raise my score if these concerns get addressed.

1. Most importantly, the writing of the paper can be **vastly** improved. I understand that the paper heavily relies on Aaron Can Den Oord et. al. (Neural discrete representation learning), and this work borrows a lot of notations from that paper. But despite this, it'd be appreciated if the authors can briefly describe the notations and succinct descriptions of key concepts (such as Babai's rounding algorithm).

2. Several choices made in this paper need to justified better? For instance why only restricting to diagonal basis for imposing structure. Several works have shown that structure can also be imposed by other structures such as a block diagonal structure, or deterministic orthonormal bases such as Hadamard frames. These bases have shown very good performance in terms of quantization reconstruction errors (although not in the context of variational autoencoders).

3. It is not really clear how the proposed strategy "effectively quantizes the data to a codebook of finite size". How can the size of the codebook be **infinite**?

4. Why is the sparsity promoting term in the loss objective used with a negative term? And even if it is used with a negative sign, wouldn't $\gamma = -1$ promote sparsity in $B$, instead of "pushing the lattice to be as dense as possible" as mentioned on Page $4$? Please correct me if I'm mistaken in my understanding.

**Questions:**

1. What is the metric for reconstruction error -- is it the Frobenius norm? Can a more perceptual metric for images be used, especially since it is hard to perceptually distinguish.

2. Can related work be moved to the introduction section. It is very difficult to appreciate the contributions of the paper otherwise.

3. Page 5: " The combined result of those two effects mentioned above " -- which two effects?

---

### Official Review · Reviewer_hAk9 · 2023-11-04

**Soundness:** 2 fair
**Presentation:** 2 fair
**Contribution:** 2 fair
**Rating:** 3
**Confidence:** 4

**Summary:**

- The authors propose “learnable lattice vector quantization,” a method for discretizing autoencoder representations in the VQ-VAE framework.
- Their method, based on Babai rounding with a diagonal basis, only requires O(D) learned quantization parameters and is faster than nearest neighbor search with a codebook.
- They control the size of the codebook with careful initialization and a “size loss term” that increases the spacing between lattice points.
- Results on FFHQ-1024, FashionMNIST, and Celeb-A show that the proposed method achieves lower reconstruction error and trains faster than VQ-VAE.

**Strengths:**

- The paper focuses on the discrete representation learning problem, an active field of research where improvements would be really useful.
- The method is described reasonably clearly.
- Results are shown on 3 different datasets, and there are visualizations of reconstructions from each model.

**Weaknesses:**

- Missing key evaluation: Method is only evaluated on reconstruction, which is not the point of discretization. Since the beginning, discretization methods have been evaluated on how they enable generative modeling as a downstream task. The foundational models in this area, VQ-VAE and VQ-VAE-2, all have results on training autoregressive models on the discretized latents. [1] trains MaskGIT, another generative model, on its discretized latents. Without these generative modeling results, it’s hard to tell whether this discretization approach is actually useful.
- I’m not convinced that the codebook size K is actually small (the paper reports K=1,405 for their method).
    - The “size loss term” (Eq. 7) $-\gamma \|diag(B)\|_1$ is not bounded below, so the entries of B should rapidly explode. This would quantize everything to 0 but minimize the total loss. The fact that this doesn’t happen makes me suspicious of the results in the paper.
    - The results in this paper could just be explained by having a very fine discretization that barely modifies the input into the quantization layer.
- Baselines: There are papers like [1] that improve the VQ training process, but this paper does not compare against them.

[1] Huh, M., Cheung, B., Agrawal, P., & Isola, P. (2023). Straightening Out the Straight-Through Estimator: Overcoming Optimization Challenges in Vector Quantized Networks. arXiv preprint arXiv:2305.08842.

**Questions:**

Major:
- How do you compute the codebook size K for your approach?
- Can you share pretrained models and code to reproduce the metrics? This would help me gain confidence in the validity and importance of the results.

Minor:
- page 1, paragraph 2:  “leveraged to train language models over continuous data (Yan et al., 2021; Bao et al., 2021).” Neither of these are about language models. The first is about video generation and the second is about representation learning via BERT-style masking.
- Should use \citep more consistently.
- Eq. 5 notation is not typical for a diagonal matrix.
- Should move the  B initialization to the main paper, since it seems like a crucial part of the method.
- Could move the architectural details in 4.1 to appendix if short on space.
- Section 4.1: What is the setting for $\gamma$?
- Section 4.1: I’m not convinced 5 epochs is enough.
- Table 2: I’m assuming duration is the training time?
- Table 2: What does No. embeddings/dataset mean? For VQ-VAE (EMA), why is it larger than the layer size?
- Page 7, where does 84/235 and 84/86 come from for the training time fraction?

---

### Author Response · Authors · 2023-11-22
**Thank you note**

We would like to thank the reviewers for their valuable feedback. It has been eye-opening and we are conducting further experiments to validate our work; however, we certainly need more time to do so. We shall therefore not be making any further revisions to our submission.

Thank you again for the time and effort you put into reviewing our work. We really appreciate it.

---

### Meta-Review · Area_Chair_z5gr · 2023-12-18

**Metareview:**

The authors propose a lattice based variant of VQ-VAEs. All reviewers have shown interest for this topic, appreciated novelty, but believe execution is still far from meeting the requirements for ICLR in terms of clarity, experimental validation, and overall quality of writing. For instance, most figures are oversized (when renormalized to their impact), and sometimes redundant (e.g. 3 and 4). The authors are encouraged to improve and "beef up" their submission.

**Justification For Why Not Higher Score:**

no rebuttal, no support

**Justification For Why Not Lower Score:**

na

---

### Decision · Program_Chairs · 2024-01-16

Reject